# Tobacco consumption behavior and its associated factors among in-school adolescent students of Saptari, Nepal

**Anil Kumar Mandal** [ORCID] *

Health and Physical Education Department, Tribhuvan University, Suryanarayan Satyanarayan Morbaita Yadav Multiple Campus, Siraha, Madhesh Province, Nepal

* anil.mandal@ssmc.tu.edu.np

## Abstract

Tobacco consumption among adolescent students in Nepal has grown to an alarming proportion, raising serious concerns about associated factors. The study aimed to describe the tobacco consumption behavior of in-school adolescent students and its associated factors. A cross-sectional survey design was followed to conduct the study. Students of community schools in Class 10 of a municipality in Saptari, Nepal, were the population of the study. All students who were present on the day of data collection participated in the study. Two hundred and twenty-five students participated in the study with a response rate of 95.74%. A validated, self-administered anonymous questionnaire was used to collect data. Frequency, percentage, and binary logistic regression were used to analyze the data. The proportion of students who consumed tobacco was 24% (boy: 18.7%, girl: 5.3%). Sex of students (OR = 5.529, 95% CI 2.375–12.872, P = .000) and tobacco consumption behavior of students' fathers (OR = 3.358, 95% CI 1.308–8.617, P = .012) were significantly associated with tobacco consumption behavior of adolescent students. I concluded that the sex of students and the tobacco consumption behavior of students' fathers are the main associated factors of the tobacco consumption behavior of adolescent students. Anti-tobacco policies should be developed and implemented with the coordination of school administration, the school management committee, and the local bodies targeting students, especially boys and students' fathers.

## Introduction

The overall prevalence of tobacco consumption among people in the world is on a decreasing trend [1], but deaths related to it are increasing around the world. By 2030, tobacco will account for 8 million deaths in the world, with an 80% contribution from low- and lower-middle-income countries due to a lack of speedy anti-tobacco programs [2]. However, another report has mentioned that 10% of adult deaths are attributed to tobacco use, and by 2030, tobacco use will take the first position to kill people, and 16.67% of deaths, or 10 million deaths, will be attributed to tobacco use, among which developing countries will contribute 7 deaths out of 10 [3]. A report [4] indicates that 21,918 people die per day and one person dies every four seconds due to tobacco use, whereas the deaths of 5 million people per

**Data availability statement:** All relevant data are within the paper and its Supporting Information files.

**Funding:** Hanumannagar Kankalini Municipality Saptari provided funds for the study on personal request. This work was supported by Hanumannagar Kankalini Municipality Saptari (Ref. 2009/078/079 to AM). There was no role of the municipality in designing, data collection, data analysis, writing the manuscript or the decision to publish.

**Competing interests:** The author has declared that no competing interests exist.

year, 13699 people per day, and one person per 6.5 seconds were attributed to tobacco use around the decade of 2000 [5]. In the context of Nepal, tobacco kills 27,137 people per year, which contributes 14.9% of all deaths in Nepal [6]. In other words, 75 people, three people, and one person die per year, per hour, and per 20 minutes due to tobacco use in Nepal. People who initiate smoking at their early adolescent stage, as more than 7 in 10 do, and carry it on for 20 years or more will shorten their lives by 20 to 25 years compared to those who never smoke [5].

Tobacco use is detrimental to health and challenging in economic aspects. In the world, more than 1.4 trillion US dollars are spent for treatment and to lose productivity due to tobacco use [2]. In India, 27.5 billion US dollars were spent for diseases attributed to tobacco use and other aspects in 2017/18, which was 1.04% of total gross domestic product, and the only treatment cost for tobacco-attributed diseases was 5.3% of total expenditures on health [7]. A diseased person who consumes tobacco spends 3% of their annual income on tobacco and its products and spends 57,215 Nepali rupees in one term of hospitalization, which is higher than the average per capita income of Nepali [8].

Although tobacco use has become a burden for health and the economy, its prevalence is higher, especially among adolescent students, which is the period of life from 10 to 19 years [9]. They enjoy risk-taking behaviors such as tobacco use behavior [10]. As a result, more adolescents from low- and middle-income countries of the world and the South-East Asian Region of the WHO consume tobacco [1]. In the world, on average, around 12%, or 43.8 million adolescents aged 12 to 15 years use at least one or more types of tobacco products. The South-East Asian countries contribute the highest number of adolescents (13–15 years) who consume tobacco. The statistics show it is 43.8 million, or 34% of the world's population of this age group; among them, 16% and 8% are boys and girls, respectively. In low- and middle-income countries, the tobacco consumption rate is higher (11–13%) among adolescents aged 13–15 years old than among adolescents of the same age in high-income countries, which is less than 10%.

In Nepal, 20.4% of adolescents consume any type of tobacco, among which 24.6% are boys and 16.4% are girls [11]. Similarly, 9% consume smoked tobacco, among which 11.4% are boys and 6.5% are girls, and 16.2% consume smokeless tobacco, among which 19.7% are boys and 12.9% are girls. Similarly, a survey study conducted by Aryal, Bista [12] in correspondence with the National Health Research Council found that 9% of adolescent students were current tobacco users, among whom 11.8% were male and 5.4% were female. It indicates that Nepal is facing a burden of tobacco because, for a state, exceeding one million SLT users or a prevalence equal to or more than 10% of SLT consumption is a situation of high burden of SLT for that state [13]. For such a condition, many factors are responsible.

Socio-economic, demographic, and spatial aspects are determinants of tobacco use among people [14]. Sex [15], age, and religion of students; education, occupation, and tobacco consumption behavior of parents; and the economic status of the family are important factors that are associated with the tobacco consumption behavior of students. The tobacco consumption behavior of adolescents is highly influenced by the tobacco consumption behavior of their parents [16,17] and other family members at home and in public places [16]. Similarly, tobacco use is higher among men, people with no education or a lower level of education, people from lower economic status, and people in higher age groups [18]. Not only this, tobacco consumption has become accepted behavior and a means of meeting with friends and relatives in Nepal [19], but also consumption of tobacco among adolescent students [16,20] is higher in districts of Madhesh Province. In this regard, this paper aims to describe the tobacco consumption behavior of in-school adolescent students and the factors associated with it.

## Methods

### Research design and research setting

This study followed a cross-sectional survey design to describe tobacco consumption behavior and its related factors among students of Class 10 from community secondary schools in Saptari, Nepal.

### Study variables

The tobacco consumption behavior of students was a dependent variable for the study. Sex, age, and religion of students; and age, education, occupation, tobacco consumption behavior of students' parents; and economic status of the family were independent variables.

### Measurement of variables

Tobacco consumption behavior of students was identified by asking whether they ever consume tobacco or not. For this study, tobacco ever consumed by students from the past to the day of data collection was taken as tobacco consumption behavior of students. Which type of tobacco product they consume with multiple responses were asked to the students to identify the types of tobacco they consumed. The tobacco consumption behavior of students' parents was measured in "yes" and "no" by asking students whether their parents consume tobacco or not. Sex of students was recorded as "male" and "female". Religion of students was categorized as "Hindu" and "Islam". The age of students and their parents was measured in continuous form, and the mean age of them is presented in the analysis. Education of parents was classified as "no formal education", and "formal education". For mother, occupation of mother was categorized into "housework", "agriculture" and "others", while occupation of father was categorized into "agriculture", "business", "foreign employment" and "others". To fulfill the assumption of binominal logistic regression [21], some categories of education and occupation of students' parents were merged. Socio-economic status of family was identified by asking students whether they think their family belongs to the poor, medium, or rich class.

### Population of the study

Students of Class 10 from all five community schools of a selected municipality of Saptari, Nepal, were the population of the study. According to the record of schools in the selected municipality, the total number of students admitted in Class 10 was 448.

### Sample size and sampling

The Yamane formula [22] was used to determine sample size at 95% confidence level, 0.5 proportion, 5% precision level, and 10% non-response rate. After calculation, it provided 232 as the number of sample sizes. Incomplete questionnaires with no response to the main dependent variables of the study were included to calculate the non-response rate in this study. To reach the targeted sample size, convenience sampling was adopted to select a municipality. All students of Class 10 of the selected municipality who were present on the day of data collection participated in the study. Altogether, 235 students participated in the study. Among them, 225 students were included in the analysis due to a response rate of 95.74%. The description of students who were admitted to each of the 5 community schools of the selected municipality and who were present on the day of study and participated in the study is presented in Table 1.

**Table 1. Number of students participated in the study from each school.**

| Schools | Students admitted in schools | | | Students presents on the day of study after excluding 10 incomplete questionnaires | | |
|---|---|---|---|---|---|---|
| | Girls | Boys | Total | Girls | Boys | Total |
| School_A | 83 | 110 | 193 | 37 | 39 | 76 |
| School_B | 23 | 24 | 47 | 12 | 10 | 22 |
| School_C | 37 | 33 | 70 | 21 | 24 | 45 |
| School_D | 64 | 55 | 119 | 38 | 28 | 66 |
| School_E | 14 | 5 | 19 | 14 | 2 | 16 |
| Total | 221 | 227 | 448 | 122 | 103 | 225 |

## Inclusion and exclusion criteria of participants

Students who were permanent residents and students of a community school of the selected municipality were included in the study. Whereas, students who had some mental or health problem, got married, were above 19 years old, and did not confirm assent paper were excluded from the study.

## Tools of data collection

An anonymous questionnaire was used to collect data that contained questions regarding the tobacco consumption behavior of students and relevant socio-demographic aspects of them and their parents. It contained mainly closed-type and a few open-type questions. The questionnaire was pretested among 15 students of Class 10 of a nearby community secondary school in the next municipality for its reliability. Some corrections were made in the questionnaire based on feedback from the pretest. Discussion with teachers and review of related literature were the means of assuring the content validity of the questionnaire.

## Data collection technique

The study was conducted from 1 July 2022 to 6 July 2022. After permission of the school administration, the purpose of the study and process of filling out questionnaire were briefly described to students. Again, students were made assured about their confidentiality and requested to provide a fair response to their best experiences. School administration managed a separate classroom for data collection, but data was collected in the absence of them to reduce bias. Students themselves filled in the anonymous questionnaire in coordination and cooperation with enumerators. After data collection, students and school administration were thanked for their coordination to complete the task.

## Data management and statistical analysis

After editing and coding, data were entered into IBM statistical package for the social sciences (SPSS) version 20 for further analysis. For the categorical variables, such as sex and religion of students, tobacco consumption behavior of students and parents, education and occupation of parents, and economic status of family frequency and percentage were used to analyze the data. For the continuous variable like age of students and parents, mean and standard deviation were calculated. Binary logistic regression [21] was applied to explain the main factors associated with the tobacco consumption behavior of students. The Enter method was applied to assess all independent variables in one block to explain the dependent variable. Statistical significance was considered if $P < .05$.

### Consent and ethical approval

Written approval was assured from the Head Teacher of selected schools for the study, and written consent from parents and students was achieved. For the study, ethical approval of the Nepal Health Research Council (Ref. No.: 4166) was confirmed.

## Results

### Demographic information of students

Table 2 shows that over half (54.2%) of the participants were girls, and the mean age of students was 15.92 (1.21) years. Most of them belonged to Hindu religion (96.4%).

### Demographic information of students' parents

Table 3 provides demographic information about students' parents. It shows that the mean age of students' fathers and students' mothers was 45.49 (7.42) and 40.38 (6.69) years, respectively. Almost all students' fathers (83.5%) had achieved formal education, whereas just over half of the students' mothers (52.9%) had achieved formal education. The main occupation of students' fathers in the study area was agriculture (49.8%), followed by business (11.4%) and foreign employment (10%), and others (28.8%). Similarly, the main occupation of students' mothers was housework (72%), followed by agriculture (19.1%). Two-thirds of students

Table 2. Demographic information of students.

| Description | | n | % |
|---|---|---|---|
| Sex of students | Female | 122 | 54.2 |
| | Male | 103 | 45.8 |
| Mean age of students | 15.92 (1.21) years | – | – |
| Religion of students | Hindu | 217 | 96.4 |
| | Islam | 8 | 3.6 |

Table 3. Demographic information of students' parents.

| Description | | n | % |
|---|---|---|---|
| Mean age of father | 45.49 (7.42) years | – | – |
| Mean age of mother | 40.38 (6.69) years | – | – |
| Education of father | Informal Education | 37 | 16.5 |
| | Formal Education | 187 | 83.5 |
| Education of mother | Informal Education | 106 | 47.1 |
| | Formal Education | 119 | 52.9 |
| Occupation of father | Agriculture | 109 | 49.8 |
| | Business | 25 | 11.4 |
| | Foreign Employment | 22 | 10.0 |
| | Others | 63 | 28.8 |
| Occupation of mother | Housework | 162 | 72.0 |
| | Agriculture | 43 | 19.1 |
| | Others | 20 | 8.9 |
| Economic status of family | Poor | 57 | 25.3 |
| | Medium | 149 | 66.2 |
| | Rich | 19 | 8.4 |

(66.2%) thought that their family belonged to the middle class, whereas 25.3% and 8.4% of students thought that their family belonged to the poor and the rich class, respectively.

### Tobacco consumption behavior of students and their parents

Table 4 explicates that among all students, 24% students consumed tobacco, and among them 18.7% were boys and 5.3% were girls. Among boys only, 40.8% boys and among girls only, 9.8% girls consumed tobacco. Supari, paan, gutkha, cigarette, and paan-parag were consumed by 75.9%, 38.9%, 33.33%, 25.9%, and 16.7% students, respectively. Likewise, 59.6% of students reported that only their father consumed tobacco, and 4.4% reported that both their father and mother consume tobacco.

### Factors associated with tobacco consumption behavior of students

Binary logistic regression [21] was performed to predict the main factors associated with the tobacco consumption behavior of students. Sex and age of students; age, occupation, education, and tobacco consumption behavior of parents; and economic status of family were included in the model as independent variables. The Omnibus Tests of Model Coefficients ($X^2$ (15, N = 225) = 65.948, P < .000) and the Hosmer and Lemeshow Test ($X^2$ (8, N = 225) = 11.412, P < .179) showed that the overall model was a good fit for analysis. The independent variables explained 26% (Cox & Snell R Square) to 38.8% (Nagelkerke R Square) variation in the dependent variable. The model correctly classified 79.5% of cases.

Table 5 reveals that the likelihood of reporting 'yes' to tobacco consumption by a boy is over five times greater (OR = 5.529, 95% CI 2.375–12.872, P = .000) than a girl reporting 'yes' to tobacco consumption. Likewise, the probability of consumption of tobacco by students whose father consumes tobacco (OR = 3.358, 95% CI 1.308–8.617, P = .012) is over three times compared to students whose parents do not consume tobacco. Having business (OR = .095, 95% CI .011–.831, P = .033) as occupation of students' father reduces the probability of tobacco consumption among students by .095 times compared to having agriculture as

Table 4. Tobacco consumption behavior of students and their parents.

| Descriptions | n | % |
|---|---|---|
| Students who consumed tobacco | 54 | 24 |
| Students who did not consume tobacco | 171 | 76 |
| Boys who consumed tobacco (among all students) | 42 | 18.7 |
| Girls who consumed tobacco (among all students) | 12 | 5.3 |
| Boys who consumed tobacco (only among boys) | 42 | 40.8 |
| Girls who consumed tobacco (only among girls) | 12 | 9.8 |
| Types of tobacco consumed by students | | |
| Gutkha | 18 | 33.3 |
| Khaini | 2 | 3.7 |
| Paan Parag | 9 | 16.7 |
| Paan | 21 | 38.9 |
| Supari | 41 | 75.9 |
| Cigarette | 14 | 25.9 |
| Others | 2 | 3.7 |
| Neither father nor mother consume tobacco | 81 | 36 |
| Both father and mother consume tobacco | 10 | 4.4 |
| Only father consume tobacco | 134 | 59.6 |

**Table 5. Factors associated with tobacco consumption behavior of students.**

| Variables | B | S.E. | Wald | df | Sig. | Exp (B) | 95% C.I. for EXP (B) | |
|---|---|---|---|---|---|---|---|---|
| | | | | | | | Lower | Upper |
| Sex of students (Female as reference) (1) | 1.710 | .431 | 15.727 | 1 | .000 | 5.529 | 2.375 | 12.872 |
| Age of students | .153 | .154 | .979 | 1 | .323 | 1.165 | .861 | 1.576 |
| Age of father | −.035 | .045 | .582 | 1 | .446 | .966 | .884 | 1.056 |
| Age of mother | .039 | .052 | .545 | 1 | .460 | 1.039 | .938 | 1.152 |
| Parental tobacco use (No parental consumption as reference) | | | 7.527 | 2 | .023 | | | |
| Parental tobacco consumption (1) | −.139 | .997 | .020 | 1 | .889 | .870 | .123 | 6.142 |
| Father tobacco consumption (2) | 1.211 | .481 | 6.346 | 1 | .012 | 3.358 | 1.308 | 8.617 |
| Occupation of mother (Housework as reference) | | | .136 | 2 | .934 | | | |
| Agriculture (1) | −.167 | .491 | .116 | 1 | .733 | .846 | .323 | 2.213 |
| Others (2) | .102 | .942 | .012 | 1 | .914 | 1.107 | .175 | 7.008 |
| Occupation of father (Agriculture as reference) | | | 8.761 | 3 | .033 | | | |
| Business (1) | −2.359 | 1.109 | 4.526 | 1 | .033 | .095 | .011 | .831 |
| Foreign employment (2) | −1.990 | 1.133 | 3.087 | 1 | .079 | .137 | .015 | 1.259 |
| Others (3) | −.897 | .480 | 3.502 | 1 | .061 | .408 | .159 | 1.043 |
| Education of father (Informal education as reference) (1) | −1.242 | .537 | 5.357 | 1 | .021 | .289 | .101 | .827 |
| Education of mother (Informal education as reference) (1) | .205 | .463 | .196 | 1 | .658 | 1.228 | .495 | 3.043 |
| Economic status (Poor as reference) | | | 1.362 | 2 | .506 | | | |
| Medium (1) | −.140 | .444 | .099 | 1 | .753 | .870 | .364 | 2.077 |
| Rich (2) | .715 | .806 | .787 | 1 | .375 | 2.045 | .421 | 9.932 |
| Constant | −3.906 | 2.934 | 1.772 | 1 | .183 | .020 | | |

occupation of students' father. Students' whose father had formal education (OR = .289, 95% CI .101–.827, P = .021) reduces .289 times the probability of tobacco consumption by students than students whose father had informal education.

## Discussion

This study aimed to describe tobacco consumption behavior of students and to find out association of different demographic aspects of students, tobacco consumption behavior of parents with tobacco consumption behavior of students. The study achieved 95.74% response rate and information of 225 students were analyzed to achieve results and conclusion. Higher number of girls participated in the study, and mean age of students was 15.92 (1.21) years. Almost all students belonged to the Hindu religion.

The mean age of students' fathers was above forty years, whereas it was forty years for students' mothers. Almost all students' fathers had access to formal education; however, just over half of the students' mothers had access to formal education; that shows the gap in access to education among men and women. Agriculture and housework were major occupation of students' fathers and students' mothers, respectively. The majority of students categorized the economic status of families as medium class. Two-thirds of students' parents consume tobacco.

The report of the Nepal Demographic Health Survey (NDHS) 2016 has mentioned that one in ten (10%) men and one-third women have no formal education [23]. Similarly, the report has mentioned that only one-third men are involved in agriculture, which is lower than the results of this study. Likewise, over two-thirds of women are engaged in agriculture occupations; that is much higher than the findings of this study. The National Planning Commission of Nepal (NPC) [24] has reported that 18.7% of the population falls below the poverty line.

Likewise, NDHS 2016 has reported that 31% of people in rural areas belong to the poorest quintiles [23]. NDHS 2016 shows that a lower percentage of men (27%) and women (6%) consumed tobacco. Due to a lack of facilities and services, people in rural areas lie below poverty line, and more people in rural areas consume tobacco. The inconsistency among the findings might be standards taken to identify poverty as another explanation for such differences.

My study showed that a higher proportion of students (one-fourth) consumed tobacco. Among boys only, tobacco consumption was much higher (one-third) than it was only among girls (one-tenth). Bhaskar, Sah [25] found 25.3% of students consumed tobacco, which is slightly higher than the findings of my study. But they reported that 31% boys only among boys and 14.4% girls only among girls consumed tobacco. For boys, it is lower, and for girls, it is higher than the findings of my study. A study [26] found a higher proportion (31.7%) of students of Classes 8 and 9 ever consumed any tobacco products, which is higher than the findings of my study. Upreti [20] found 22.8% of adolescent students consumed tobacco, which is slightly lower than the findings of my study. Next studies show that prevalence of tobacco consumption among students was 15.6% [16] and 19.7% [27], which is lower than findings of my study. Similarly, they found a lower proportion of boys and girls who consumed tobacco among boys only and among girls only, respectively, than the findings of my study. Pradhan, Niraula [27] reported 33.6% boys and 4% girls consumed tobacco, and Chaudhary and Bhandari [16] found that 29.1% boys and 1.3% girls consumed tobacco. The higher proportion of students who consumed tobacco in this study might be due to the fact that most of the area of this municipality is rural, and the prevalence of tobacco consumption among students was higher who were from rural areas (23.5%) than among students who were from urban areas (21.9%) [28]. Findings of my study support the social role of people. Tobacco consumption in our society is considered a function of men, whereas it is considered a shame for women. This is why more boys and less girls consume tobacco [29]. However, this study reveals that consumption of tobacco among girls is also rising and cannot be neglected.

Consumption of Supari was the most popular among students, followed by Paan, Gutkha, and cigarettes. Joshi, Pradhan [30] also found betel nuts (40.5%) as the most popular among student but they found only 6% of students consume smokeless tobacco. Unlike this study, cigarette was popular among students (44.4%) than paan-parag (25%) and gutkha (11.2%) [20]. Similarly, Chaudhary and Bhandari [16] found that more than two-thirds (70.8%) of students smoked and others consumed smokeless tobacco. However, Bhaskar, Sah [25] reported that only 7.7% of students smoked and 17.6% of students consumed smokeless tobacco, whereas Pradhan, Niraula [27] found that among tobacco users, 17.9% and 8% of students ever consumed cigarette and smokeless tobacco, respectively. Differences are found among the studies. Easy availability and accessibility of Supari (betel nuts) at our home might be an explanation for why more students consume Supari.

This study found that the sex of students is a significant factor that determines the tobacco consumption behavior of students, but the age of the student is not a significant factor that relates to tobacco consumption behavior of the students. My study reveals that being a boy increases the probability of tobacco consumption more than five times than being a girl. Being a boy student increased consumption of smoked tobacco and smokeless tobacco six times and seven times, respectively, than being a girl student [26]. Study of Chaudhary and Bhandari [16] partially supports findings of this study. They reported a significant association between sex and age of students and tobacco consumption behavior of them, Another study also found that sex of students significantly associated with tobacco consumption behavior of them [25]. They found that being a boy raised the probability of tobacco consumption more than twofold (OR = 2.65) among students, which is lower than the findings of my study. Likewise, a

systematic study [31] reveals that sex and increasing age are significant factors that determine the tobacco consumption behavior of adolescent students. The theory of triadic influence [32] considers sex of students as one of the most important individual-level components that determines their behavior, like tobacco consumption. The finding of this study is in line with this theory.

This study showed the probability of consumption of tobacco by students whose fathers had formal education was .289 times lower compared to students whose fathers had informal education. Like the findings of my study, a study [15] conducted in Syangja, Nepal, among students found a significant association between tobacco consumption behavior of students and their father education. The prevalence of tobacco consumption was significantly lower among students having parents with formal education than among students having parents with no formal education because higher education of the father is one of the protective factors of tobacco use [31]. The educated people are aware to dangerous effects of tobacco use, so they prevent themselves and their children from consumption of it.

The present study found that the likelihood of consumption of tobacco by students whose fathers were in business was .095 times lower than students whose fathers were involved in agriculture occupation. A study conducted in Dharan found a significant association between tobacco consumption behavior among students and the profession of their father [27]. They found that the probability of consumption of tobacco among students whose fathers do skilled or semiskilled work in a foreign country was higher than students whose fathers were in the service of a professional job. It is also supported by the report of NDHS [23] that reported people from professions like agriculture consume more tobacco. Most of the students' parents in this study were involved in agriculture (Table 3) and consumed tobacco (Table 4), which raised the probability of tobacco consumption among the students.

This study highlighted that the odds of tobacco consumption among adolescent students whose fathers consumed tobacco were more than three times higher than whose fathers did not consume tobacco. A study conducted among adolescents of age 10–19 years found that smoking was more than two times (AOR = 2.57) higher among adolescents whose parents smoked than whose parents did not smoke [33]. Likewise, many studies have found similar findings to this study. Tobacco consumption behavior of students was significantly influenced by tobacco consumer parents [16,32,34] and relatives [35]. The tobacco consumption behavior of parents is significant predictor of the tobacco consumption behavior of their adolescent children [31]. Similarly, the tobacco consumption behavior of family members significantly increased the likelihood of tobacco consumption among students 11 folds compared to students whose family members did not consume tobacco [25]. Children are very close to their parents and other family members. Hence, they have enough time to observe the behavior of their parents and other family members. Observation of the tobacco consumption behavior of parents and other family members might raise the probability of tobacco consumption by children because observation is one of the most important means of learning [36].

## Limitation of the study

This study was based on a cross-sectional design, and data was collected by using self-administered questionnaire; therefore, there might be the probability of over- or under estimated or both data. Questions regarding generalization might arise as this study only included students from community schools in only one municipality. This study only included socio-economic, demographic, spatial, and behavioral aspects of parents as predictor variables. Besides these limitations, students were appreciated for providing factual information according to their experience to maintain the quality of the data.

## Conclusion

My study reveals that a higher proportion of adolescent students, especially boys, consume tobacco. Consumption of Supari, Paan, and Gutkha is common among students. The sex of students and the tobacco consumption behavior of students' fathers are the main factors that are associated with the tobacco consumption of students. Besides it, the occupation and education of father are also significant factors related to the tobacco consumption behavior of students. To minimize tobacco consumption among students, the school administration should totally prohibit tobacco consumption at school premises; an anti-tobacco program should be conducted by the school management committee for students and parents at least once a month; and anti-tobacco policies for students, especially for boys, should be developed and implemented in coordination with the school administration, the school management committee, and the local bodies.

## Supporting information

**S1 Data.  Data set of article.**
(XLSX)

## Acknowledgments

I would like to heartily thank the team of Hanumannagar Kankalini Municipality, Saptari, especially Mr. Surendra Kumar Yadav, former chief executive officer, for his coordination in the municipality for economic support. Similarly, my thanks go to school administration and teachers for their coordination, students' parents for their consent, and students who actively participated in the study. I remember the enumerators who collected the data and coordinated for the study. At last but not least, I am grateful to Mr. Chun Chun Yadav, who cooperated with me during study and edited the language of the manuscript, and Mr. Bhupendra Prasad Yadav, who cooperated with me during data collection procedures in the schools.

## Author contributions

**Conceptualization:** Anil Kumar Mandal.

**Data curation:** Anil Kumar Mandal.

**Formal analysis:** Anil Kumar Mandal.

**Funding acquisition:** Anil Kumar Mandal.

**Investigation:** Anil Kumar Mandal.

**Methodology:** Anil Kumar Mandal.

**Project administration:** Anil Kumar Mandal.

**Resources:** Anil Kumar Mandal.

**Software:** Anil Kumar Mandal.

**Supervision:** Anil Kumar Mandal.

**Validation:** Anil Kumar Mandal.

**Visualization:** Anil Kumar Mandal.

**Writing – original draft:** Anil Kumar Mandal.

**Writing – review & editing:** Anil Kumar Mandal.

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
