## [Decision Letter · Decision Letter 0]

15 Jul 2024

PGPH-D-24-01222

Factors associated with tobacco consumption behavior of adolescent students

Dear Dr. Mandal, 

Thank you for submitting your manuscript to PLOS Global Public Health. After careful consideration, we feel that it has merit but does not fully meet PLOS Global Public Health’s publication criteria as it currently stands. Therefore, we invite you to submit a revised version of the manuscript that addresses the points raised during the review process.

We look forward to receiving your revised manuscript.

Kind regards,

Nnodimele Onuigbo Atulomah, PhD

Academic Editor

Journal Requirements:

1. We are unable to open your Supporting Information file "Data set of Article in SPSS 20.sav". Please kindly revise as necessary and re-upload.

2. In the online submission form, you indicated that [Insert text from online submission form here]. 

3. Uploaded as supplementary information.

Additional Editor Comments (if provided):

The reviewers have made their observations and recommendations for revision. Kindly consider carefully the comments and observations of the reviewers before implementing the suggested revisions. As the academic editor I have observed weaknesses in the manuscript that would greatly benefit from suggested modifications:

TITLE: Definitely, the title of the study would required attention. The title appear incomplete because it has not identified the location of the study.

ABSTRACT: The abstract needs extensive native English editorial corrections. For example, the first paragraph of the abstract should read: “Tobacco consumption of adolescent students in Nepal have consistently grown to an alarming proportion raising serious concerns of associated factors for the observed trends. This study sought to identify and characterize factors associated with tobacco consumption behavior of these adolescent students.”

The abstract is grossly inadequate because the factors of importance and their association with the outcome variable are missing. Therefore, in what way would it be said that this abstract represents the study?

Consider appropriate keywords related to MeSh-(Medical Subject Headings).

INTRODUCTION: The introduction lacked certain theoretical principles to be resolved based on the statement in lines 62-66. The statement leaves a serious theoretical gap necessary to elucidate observed tobacco consumption among this population. Note that the study is not merely stating the obvious but scientifically substantiating such statement. Without linking this all-important observation to a valid theoretical framework weakens its foundation. Furthermore, how may the observations in lines 238-243 of the study be explained by the existing theoretical framework, apparently there is none, therefore in the absence of any scientifically grounded explanations, the study remains weak, and speculative.

METHODS: The section of the methodology needs to be revised to demonstrate precision. For example; line 75: “A cross-sectional survey research design, a non-experimental research where questionnaire or interview is the means of data collection to know the features of population through sample..., was applied for the study.” should be written as “The study adopted a cross-sectional survey design for data collection from consenting adolescents attending the community secondary schools at Hanumannagar Kankalini Municipality, Saptari”.

How were the study variables determined without omitting vital other variables. The categorization of the variables appear inappropriate.

The measurement strategies for the behaviour variable is faulty. A re-analysis is required.

Reviewers' comments:

Reviewer's Responses to Questions

**Comments to the Author**

1. Does this manuscript meet PLOS Global Public Health’s publication criteria ? Is the manuscript technically sound, and do the data support the conclusions? The manuscript must describe methodologically and ethically rigorous research with conclusions that are appropriately drawn based on the data presented.

Reviewer #1: Yes

Reviewer #2: Yes

Reviewer #3: Partly

2. Has the statistical analysis been performed appropriately and rigorously?

Reviewer #1: Yes

Reviewer #2: No

Reviewer #3: No

3. Have the authors made all data underlying the findings in their manuscript fully available (please refer to the Data Availability Statement at the start of the manuscript PDF file)?

Reviewer #1: Yes

Reviewer #2: Yes

Reviewer #3: No

4. Is the manuscript presented in an intelligible fashion and written in standard English?

Reviewer #1: No

Reviewer #2: Yes

Reviewer #3: No

5. Review Comments to the Author

Reviewer #1: You have conducted a very important and timely study. However, there is a need to present the manuscript in a correct and clear English language; you can seek help with some soft applications. From the first sentence in the introduction to almost the entire manuscript, the language lacks clarity, and some sentences are incomplete, some are repetitions.

The scope of the study is not completely captured in the title and I suggest the title to be "Tobacco consumption behaviour and associated factors among adolescents", since the study delved deeply into consumption behaviour, in addition to factors associated with them.

Results are grossly repeated in the section. Please research what should be contained in the discussion section of a quantitative study. When attempts were made to compare your findings with those of previous studies, they were not properly written and it was difficult to understand the relationship between your findings and those of the cited literature.

Thank you

Reviewer #2: This manuscript presents an investigation to the factors associated with tobacco consumption behavior of adolescent. The manuscript is generally fairly written and clearly presented.

Other general and specific comments are provided below:

1. Originality of value: The manuscript presents an original study which was designed and conducted among adolescent students in Nepal. The paper addressed a relevant public health issue of concern, namely tobacco consumption with interesting results.

2. Suitability and soundness of technique: The techniques used for the study are adequate and robust. However, the inferential statistics is inadequate the authors should have utilized binary logistic regression alongside with the chi-square analysis.

3. Clarity of Presentation: The manuscript was fair in narrative presentation. It is easy to follow and complied with the journal specifications.

4. Areas requiring correction. The paper is generally fairly written, there are areas I will like the authors to correct;

Title: The title should be change to reflect the study area. Suggested title: Factors associated with tobacco consumption among in-school adolescents in Nepal

Abstract

Line 2 on the abstract should not start with although since the consumption of tobacco among the adolescents in Nepal is still a problem.

Line 4: the objective of the paper should be written in past tense since authors were reporting their findings.

Line 5: the design should simply be written as cross-section design or survey design.

Line 7; the authors should change probability proportional to proportional stratified sampling technique.

Line 8 instead of stating that the sample size was 235 but only 225 were included in analysis, the authors should state that the response rate was 95.7%.

Line 9: the authors need to include that the questionnaire was validated.

Line 12: the statistics value to show the association should be included in this abstract.

Line 15: the recommendation should be change to be all encompassing not just focusing on the girls and who should be responsible for the recommendation made has to be specified.

Line 17 Keywords- factor should change as keywords it can be replaced with parent consumption.

Introduction

Line 19 and 26 is stating the same thing on the number of death attributed to tobacco. The authors should be mindful of repeating the same thing.

The author should clearly state the gap and the uniqueness of the study as the factors were already reported by a study as noted in line 62.

Methods

Line 75 the authors should state that the study employed a cross-sectional design.

Line 102, the authors should explain the basis for different non-response rate.

Line 103: the authors should explain in details the sampling technique.

How many boys were selected per schools and how many girls? It will be best to have a table to show this distribution and how did they select the students that fill the questionnaire.

Line 114: The authors should include where the pretest of instrument was done and the type of validation that was done in details.

Results

Line 155: the authors should create a table that shows boys and girls separate instead of lumping them together.

Line 156: the authors should use a binary logistic regression, Chi-square is not sufficient.

Line 256: the recommendation is weak and it should be all encompassing.

Reviewer #3: Thank you for inviting me to review this manuscript. The following are my contributions in the field of the subject area.

The title syntax requires important modification to enable it to communicate its focus appropriately. The area of focus should be included in the title. Suggestion” Factors associated with tobacco consumption behavior of adolescent student in Saptari, Nepal.

Abstract: The abstract requires editorial re-structuring to make it precise. Line 2 should not begin with “although” The description of the methodological approach is not well articulated. Line 5-8 should read thus, “ Cross-sectional survey design was utilized in this study. A sample size of 235 students from community class 10 of Hanumannagar kankalini Municipality, saptari was determined using probability proportion to size method. Statement of conclusion require clarity.

Introduction: The introduction requires English language editorials.

Line 19-21 can be rephased for clarity of purpose. The deceleration of tobacco cessation program will attribute to 8 million deaths by 2030, where 80% of death will occur in low and middle-income countries.

Methods: The statement of the study made in line 75 would be more appropriately expressed: thus The study conducted was a cross sectional survey design to determine Factors associated with tobacco consumption behavior of adolescent student in Saptari, Nepal.

The instrument for data collection should be clearly stated. Was the instrument for data collection a questionnaire or interview or both?

Tools for data collection; Data collection needs to be detailed. The number of items on the questionnaire needs to be stated and scored.

Note; The acronym SPSS should be written in full and line 124 should be corrected “Data analysis was conducted with IBM Statistical Product and Services Solution (SPSS version 20)”

Conclusion; Data analysis is flawed and as such discussion of the implications of data attempting to explain the dynamics of the problem phenomenon will be weak and flawed as well.

6. PLOS authors have the option to publish the peer review history of their article (what does this mean? ). If published, this will include your full peer review and any attached files.

**Do you want your identity to be public for this peer review?** For information about this choice, including consent withdrawal, please see our Privacy Policy .

Reviewer #1: **Yes: ** Ukamaka Gladys Okafor

Reviewer #2: **Yes: ** Titilayo Olaoye

Reviewer #3: No

---

## [Decision Letter · Decision Letter 1]

20 Sep 2024

PGPH-D-24-01222R1

Tobacco consumption behavior and its associated factors among adolescent students of Saptari, Nepal

Dear Dr. Mandal,

Thank you for submitting your manuscript to PLOS Global Public Health. After careful consideration, we feel that it has merit but does not fully meet PLOS Global Public Health’s publication criteria as it currently stands. Therefore, we invite you to submit a revised version of the manuscript that addresses the points raised during the review process.

The manuscript has been evaluated by two reviewers, and their comments are available below. Reviewer 2 has raised a concern that a binary linear regression analysis is more suitable for the analysis of your data than a Chi Square test. Please could you reanalyze your data using a binary linear regression analysis, to ensure that the experiments, statistics, and other analyses are performed to a high technical standard and please ensure that they are described in sufficient detail, as required in our publication criteria (https://journals.plos.org/globalpublichealth/s/criteria-for-publication). Thank you very much for your attention to this. 

We look forward to receiving your revised manuscript.

Kind regards,

Johanna Pruller, Ph.D.

PLOS Staff Editor

Additional Editor Comments (if provided):

Reviewers' comments:

Reviewer's Responses to Questions

**Comments to the Author**

1. If the authors have adequately addressed your comments raised in a previous round of review and you feel that this manuscript is now acceptable for publication, you may indicate that here to bypass the “Comments to the Author” section, enter your conflict of interest statement in the “Confidential to Editor” section, and submit your "Accept" recommendation.

Reviewer #1: All comments have been addressed

Reviewer #2: (No Response)

2. Does this manuscript meet PLOS Global Public Health’s publication criteria ? Is the manuscript technically sound, and do the data support the conclusions? The manuscript must describe methodologically and ethically rigorous research with conclusions that are appropriately drawn based on the data presented.

Reviewer #1: Yes

Reviewer #2: Yes

3. Has the statistical analysis been performed appropriately and rigorously?

Reviewer #1: Yes

Reviewer #2: No

4. Have the authors made all data underlying the findings in their manuscript fully available (please refer to the Data Availability Statement at the start of the manuscript PDF file)?

Reviewer #1: Yes

Reviewer #2: Yes

5. Is the manuscript presented in an intelligible fashion and written in standard English?

Reviewer #1: Yes

Reviewer #2: Yes

6. Review Comments to the Author

Reviewer #1: Author has addressed the major issues raised by the reviewers

Reviewer #2: Review uploaded

7. PLOS authors have the option to publish the peer review history of their article (what does this mean? ). If published, this will include your full peer review and any attached files.

**Do you want your identity to be public for this peer review?** For information about this choice, including consent withdrawal, please see our Privacy Policy .

Reviewer #1: **Yes: ** Ukamaka Gladys Okafor

Reviewer #2: **Yes: ** Titilayo Olaoye

---

## [Decision Letter · Decision Letter 2]

17 Dec 2024

Tobacco consumption behavior and its associated factors among in-school adolescent students of Saptari, Nepal

PGPH-D-24-01222R2

Dear Mr. Mandal,

We are pleased to inform you that your manuscript 'Tobacco consumption behavior and its associated factors among in-school adolescent students of Saptari, Nepal' has been provisionally accepted for publication in PLOS Global Public Health.

Best regards,

Julia Robinson

Executive Editor

Reviewer Comments (if any, and for reference):

Reviewer's Responses to Questions

**Comments to the Author**

1. If the authors have adequately addressed your comments raised in a previous round of review and you feel that this manuscript is now acceptable for publication, you may indicate that here to bypass the “Comments to the Author” section, enter your conflict of interest statement in the “Confidential to Editor” section, and submit your "Accept" recommendation.

Reviewer #1: All comments have been addressed

2. Does this manuscript meet PLOS Global Public Health’s publication criteria ? Is the manuscript technically sound, and do the data support the conclusions? The manuscript must describe methodologically and ethically rigorous research with conclusions that are appropriately drawn based on the data presented.

Reviewer #1: Yes

3. Has the statistical analysis been performed appropriately and rigorously?

Reviewer #1: Yes

4. Have the authors made all data underlying the findings in their manuscript fully available (please refer to the Data Availability Statement at the start of the manuscript PDF file)?

Reviewer #1: Yes

5. Is the manuscript presented in an intelligible fashion and written in standard English?

Reviewer #1: Yes

6. Review Comments to the Author

Reviewer #1: (No Response)

7. PLOS authors have the option to publish the peer review history of their article (what does this mean? ). If published, this will include your full peer review and any attached files.

**Do you want your identity to be public for this peer review?** For information about this choice, including consent withdrawal, please see our Privacy Policy .

Reviewer #1: **Yes: ** Ukamaka Okafor
